# Arabic validation and cross-cultural adaptation of the 5C scale for assessment of COVID-19 vaccines psychological antecedents

Samar Abd ElHafeez[1], Iffat Elbarazi[2]*, Ramy Shaaban[3], Rony ElMakhzangy[4], Maged Ossama Aly[5], Amr Alnagar[6], Mohamed Yacoub[7], Haider M. El Saeh[8], Nashwa Eltaweel[9], Sulafa T. Alqutub[10], Ramy Mohamed Ghazy[11]

1 Epidemiology Department High Institute of Public Health, Alexandria University, Alexandria, Egypt, 2 Institute of Public Health, College of Medicine and Health Sciences, United Arab Emirates University, Al Ain, Abu Dhabi, United Arab Emirates, 3 Department of Instructional Technology and Learning Sciences, Utah State University, Salt Lake City, Utah, United States of America, 4 Alexandria University, Alexandria, Egypt, 5 High Institute of Public Health, Alexandria University, Alexandria, Egypt, 6 General Surgery Department, Faculty of Medicine, Alexandria university, Alexandria, Egypt, 7 Department of English, Florida International University, Gainesville, Florida, United States of America, 8 Department of Community Medicine, Faculty of Medicine, University of Tripoli, Tripoli, Libya, 9 Obstetrics and Gynaecology Department, University Hospitals of Birmingham, Birmingham, England, United Kingdom, 10 Family and Community Medicine Department, Faculty of Medicine, University of Jeddah, Jeddah, Kingdom of Saudi Arabia, 11 Tropical Health Department, High Institute of Public health, Alexandria University, Alexandria, Egypt

* ielbarazi@uaeu.ac.ae

**Data Availability Statement:** Data used in this study are available from the OSF database at

## Abstract

### Background

In the Arab countries, there has not been yet a specific validated Arabic questionnaire that can assess the psychological antecedents of COVID-19 vaccine among the general population. This study, therefore, aimed to translate, culturally adapt, and validate the 5C scale into the Arabic language.

### Methods

The 5C scale was translated into Arabic by two independent bilingual co-authors, and then translated back into English. After reconciling translation disparities, the final Arabic questionnaire was disseminated into four randomly selected Arabic countries (Egypt, Libya, United Arab Emirates (UAE), and Saudi Arabia). Data from 350 Arabic speaking adults (aged ≥18 years) were included in the final analysis. Internal consistency was assessed by Cronbach's alpha. Construct validity was determined by concurrent, convergent, discriminant, exploratory and confirmatory factor analyses.

### Results

Age of participants ranged between 18 to 73 years; 57.14% were females, 37.43% from Egypt, 36.86%, from UAE, 30% were healthcare workers, and 42.8% had the intention to get COVID-19 vaccines. The 5 sub-scales of the questionnaire met the criterion of internal consistency (Cronbach's alpha ≥0.7). The predictors of intention to get COVID-19 vaccines

https://osf.io/hukre [or https://osf.io/46h8z/] (DOI:
10.17605/OSF.IO/46H8Z).

**Funding:** The author(s) received no specific
funding.

**Competing interests:** The authors have declared
that no competing interests exist.

(concurrent validity) were young age and the 5C sub-scales. Convergent validity was identified by the significant inter-item and item-mean score of the sub-scale correlation ($P<0.001$). Discriminant validity was reported as inter-factor correlation matrix (<0.7). Kaiser-Meyer-Olkin sampling adequacy measure was 0.80 and Bartlett's sphericity test was highly significant (*P<0.001*). Exploratory factor analysis indicated that the 15 items of the questionnaire could be summarized into five factors. Confirmatory factor analysis confirmed that the hypothesized five-factor model of the 15-item questionnaire was satisfied with adequate psychometric properties and fit with observed data (RMSEA = 0.060, GFI = 0.924, CFI = 0.957, TLI = 0.937, SRMR = 0.076 & NFI = 906).

## Conclusion

The Arabic version of the 5C scale is a valid and reliable tool to assess the psychological antecedents of COVID-19 vaccine among Arab population.

## Introduction

The world is currently in a public health crisis facing a fierce virus, the coronavirus disease (COVID-19), which puts the world in a pandemic [1]. Till the mid of May, 2021, there was more than 165 million reported COVID-19 cases and more than 3 million deaths worldwide. Among them, about 8 million cases and almost half million deaths in the Arab world [2].

All countries around the world are fighting the spread of COVID-19. Procedures that countries have taken include enforcing quarantines, lockdowns, social distancing, wearing facemasks, and travel restrictions. These procedures have affected people both physically and psychosocially and have massively left negative impacts on the global economy. "The multifaceted catastrophic consequences associated with the COVID-19 outbreak have intensified international efforts in developing an effective prevention method to keep outbreaks under control" [3].

A combined effort is being simultaneously exerted by the World Health Organization (WHO), international governmental sectors, academic communities, and pharmaceutical industries to develop and deploy safe and effective vaccines. As of 21 May, 2021, there are 184 vaccines are in a pre-clinical development phase, with 100 vaccines selected to reach the clinical development stage [4]. The WHO listed Sinopharm, Pfizer/BioNTech, Astrazeneca-SK Bio, Serum Institute of India, and Janssen and Moderna vaccines for emergency use [5]. The COVID-19 Vaccines Global Access (COVAX) is a global coalition that aims to roll out equitable distribution of the vaccines to all countries and to ensure that vulnerable populations are high priorities. The Covid-19 vaccines are available in all Arab countries. The most commonly used COVID-19 vaccines are Pfizer Biontech, Sinopharm, AstraZeneca, and Sputnik [6]. Till 21th of May, more than 50 million vaccine doses were administered in the Arab league countries, with 11.7 million of doses administered in the UAE and only 2500 doses in Syria [7].

The production of an effective vaccine against COVID-19 virus faces several challenges such as selecting a proper formulation, reviewing and approving a large number of potential vaccine candidates, massively producing the vaccine, and surveilling it in the post-marketing stage, cost issues and logistics of distribution [8–10]. Nevertheless, a major obstacle towards achieving appropriate vaccination and reaching an eventual herd immunity can be vaccine hesitancy among the general public. Newly emerging vaccines are usually questioned by

community members and the views on receiving them can vary dramatically between individuals [11].

The Strategic Advisory Group of Experts (SAGE) on Immunization concluded that vaccine hesitancy (VH) refers to *delay in acceptance or refusal of vaccines despite availability of vaccine service*. The SAGE reported that VH is influenced by several factors as complacency, convenience, and confidence [12]. Vaccine hesitancy describes a continuum between complete acceptance and complete refusal, which could slow the fight against COVID-19 infection[10].

COVID-19 vaccine acceptance is context-specific and varies with geography, culture, and sociodemographic. In a global survey conducted by Lazarus V *et al* [13]; 71.5% responded that they would accept to take the vaccine in case it was proven safe and effective, and 48.1% said that they would get vaccinated if their employer suggested it. More than 70% of 7662 participants from seven European countries demonstrated their willingness to get vaccinated against COVID-19 infection [14]. A worldwide systematic review on COVID-19 vaccine acceptance reported that the highest acceptance rates of COVID-19 vaccination were in Ecuador (97.0%), Malaysia (94.3%), Indonesia (93.3%) and China (91.3%). On the opposite side, the lowest acceptance rates of COVID-19 vaccination were in Kuwait (23.6%), Jordan (28.4%) [15]. The low rates of vaccine acceptance could be returned to the widespread embrace of conspiratorial beliefs in the Arab region, with its subsequent negative attitude towards vaccination [16–18]. Other published studies in Arab nations showed that COVID-19 vaccination acceptance rate varies between 29.4%- 64.7% [19–24].

There are several tools that have been developed and validated for assessment of vaccine acceptance and hesitancy. Some of them include; Vaccine Confidence Scale [25], Parent Attitudes about Childhood Vaccines Survey [26], Vaccine Hesitancy Scale (VHS) [27], Global Vaccine Confidence Index [28], and the 5C scale [29].

Betsch et al., (2018) [29] developed and validated the 5C scale to assess the VH towards vaccine preventable diseases among the German and American populations. This tool widens the scope of the measures and the theoretical and conceptual frameworks used to study VH and acceptance. All available tools depend on 3C model (confidence, complacency, constraints) to assess the VH. The 5C scale provides more in-depth understanding of the "individual mental representations, attitudinal and behavioral tendencies that are a result of the environment and context the respondent lives in". It assesses five psychological determinants pertaining to the individual's vaccination decision: confidence, complacency, constraints, calculation, and collective responsibility [30]. As a limitation, the 5C scale authors have pointed out the difficulty of generalizing the predictive validity of the 5C tool if not tested in other countries and on other populations [29].

Identifying the population acceptance of the COVID-19 vaccine necessitates the use of validated tools to reflect the real picture. No published article in the Arab world has yet reported data on any valid tool to assess COVID-19 vaccine hesitancy. This impacts the validity of the findings and explicates the gap in COVID-19 vaccine acceptance among different countries and populations in the region. This study, therefore, aims to translate, culturally adapt, and validate the 5C questionnaire into the Arabic language to be used as a standardized tool for assessment of the psychological antecedents against COVID-19 vaccination in the Arab region and allow for comparison of VH rates across different countries.

## Methods

### Study design and setting

A cross sectional survey method was used. This study is part of a large multi-national project to assess the psychological antecedents against COVID-19 vaccines among Arab populations

living in 14 Arab countries (Egypt, Sudan, Libya, Tunisia, Morocco, Mauritania, Jordan, Palestine, Lebanon, United Arab of Emirates (UAE), Saudi Arabia, Oman, Kuwait, and Yemen). Four randomly selected Arab countries were included in the current study. A representative researcher from each selected country was assigned to collect data from that country.

## Data collection tool

A survey of two sections was distributed to collect the data. **The first section** included questions on sociodemographic data (age, sex, country, nationality, education, marital status, and healthcare profession), history of COVID-19 infection, history of relatives' death due to COVID-19 infection, knowledge about the availability of different types of COVID-19 vaccines, and intention to get COVID-19 vaccine. **The second section** included a translated version of the 15-item 5C scale to assess the psychological antecedents to COVID-19 vaccination (S1 and S2 Tables). It covers five sub-scales; *confidence* which means trust in the effectiveness and safety of vaccines, or the system that delivers them, including the reliability and competence of the health services and health professionals, and also the motivations of policy-makers who decide on the need of vaccines, *complacency* that refers to the existence of low perceived risks of vaccine-preventable diseases and so vaccination is not deemed a necessary preventive action, *constraints* related to the physical availability, affordability and willingness-to-pay, geographical accessibility, ability to understand (language and health literacy) and appeal of immunization service, *calculation* which alludes the individuals' engagement in extensive information searching on the with perceived vaccination and disease risks, and *collective responsibility* that conveys the willingness to protect others by one's own vaccination by means of herd immunity [31].

**Score interpretation.** Each of the 5 sub-scales (confidence, complacency, constraints, calculation, and collective responsibility), was assessed by 3 rating items on a 7-point scale (1 = strongly disagree, 2 = moderately disagree, 3 = slightly disagree, 4 = neutral, 5 = slightly agree, 6 = moderately agree, 7 = strongly agree). The mean scores of items under each sub-scale were computed, with higher mean score indicating stronger agreement of the corresponding sub-scale. Using the 5C scale does not lead to a total score providing a sample's absolute state of hesitancy. It, rather, allows for a valid assessment of the different psychological antecedents [29, 30].

**Translation and adaptation.** This step was done by six of this manuscript authors and one certified Arabic translator. We forward-translated the 5C scale into formal Arabic by two independent bilingual co-authors (AA & NE). Both co-authors rated the difficulty of translating each item and the associated response choices. One bilingual researcher (RS) and another Arabic translator compared the two translations and reconciled the discrepancies. Then, the questionnaire was back translated into English by two additional co-authors (MY & RE). The back translators with the first author (SA) compared their translations with the previous English version. Minor discrepancies were identified and resolved by discussions between the researchers.

**Content validity and expert evaluation.** The next step in the validation process was to assess the content validity with an expert panel of 10 investigators (methodologist, healthcare professionals, public health professional, and language professionals). The expert panel examined whether the agreed-on translation covers the concepts as defined. In addition, all researchers (48 researchers) from the 14 Arab countries were invited to revise the Arabic copy of the 5 C questionnaire and give their feedback.

**Pilot testing and cognitive interviews.** We next performed cognitive testing of the Pre-final version. Trained members of the research team conducted cognitive interviews among 20

participants of the intended respondents (5 from each included country) to evaluate participants' understanding, readability, language, wording, and cultural appropriateness of items as well as the clarity of the instructions for providing responses for each section.

During this step, we encountered some difficulties with explaining some points. The first comment was related to the seven points Likert scale, particularly the difference between strongly agree/disagree and moderately agree/disagree. In the Arabic language, there is no sharp demarcation between the perceived meaning of strongly and moderately. Another item, which was not well understood by the participants, is the "Everyday stress prevents me from getting vaccinated", there was a confusion regarding the real perspective of the daily stress that will hinder them from taking the vaccine. Some participants felt that there was a repetition of the questions "Vaccination is unnecessary because vaccine-preventable diseases are not common anymore" and "Vaccine-preventable diseases are not so severe that I should get vaccinated". Also, the question "For me, it is inconvenient to receive vaccinations". Some participants were unable to define the precise meaning of inconvenience and how inconvenience would impact their ability to consider the vaccine. We reformulated the Arabic questions to deliver the construct beyond each item of the original copy of the questionnaire. Then, the final Arabic version was approved by the researchers and was ready for field-testing.

## Sample size for testing the validity of the Arabic version of 5C scale

Based on the sample size recommendations of having 10 participants respond to each item for validating a questionnaire (ratio 10:1), we needed 150 participants [32]. Moreover, a priori sample size calculation for Structural Equation Modelling (SEM) technique to perform confirmatory factor analysis (CFA) showed that a minimum sample of 200 is required to run CFA [33]. For that, the minimum required sample size for our analysis was 350 participants. Adult (18 years and above) who are Arabic speaking from the Arab countries included in the study.

**Sampling technique and data sources.** The final Arabic copy of 5C scale (S2 Table) was uploaded on Qualtrics and disseminated online via different social media platforms (Facebook, WhatsApp, emails, and Twitter) to 673 participants from December 14, 2020 until January 14th, 2021. The sample was recruited from the four randomly selected Arabic countries (Egypt, Saudi Arabia. Libya and United Arab of Emirates (UAE). Each representative researcher was responsible for submitting the questionnaire to the social media platform groups from his country. The latest digital data reported that Internet penetration was at 99%, 95.7%, 75%, and 57.3%, and the number of social media users was equivalent to 99%, 79.3%, 75%, and 47.3% of the total population living in UAE, Saudi Arabia, Libya, and Egypt [34]. A total of 511 responded to the questionnaire, 89 participants chose not to complete the questionnaire. The response rate was 62.70% (422/673). Of the 422 who completed the questionnaire, we excluded 72 responses from the final analysis due to incomplete or inconsistent data (33 from Egypt, 16 from Libya, 12 from Saudi Arabia, and 11 from UAE). The final sample size included in our analysis was 350 participants Fig 1. Participants completed the survey after reading a clearly developed information that explained the purpose and nature of the study, the privacy and confidentiality of the data, and that the participation was voluntary, and no financial compensation would be provided. Only those who clicked I agree to participate were able to initiate the questionnaire via Qualtrics.

## Data management and psychometric analysis

Quantitative variables are summarized as mean ± standard deviation (SD) while qualitative variables are presented with percent and frequency. Mean scores of each sub-scale were calculated. Pearson's correlation analysis was used to calculate inter-item and item- to- mean score of the sub-scale correlation. Multiple logistic regression analysis was used to calculate the odds ratio (OR) and

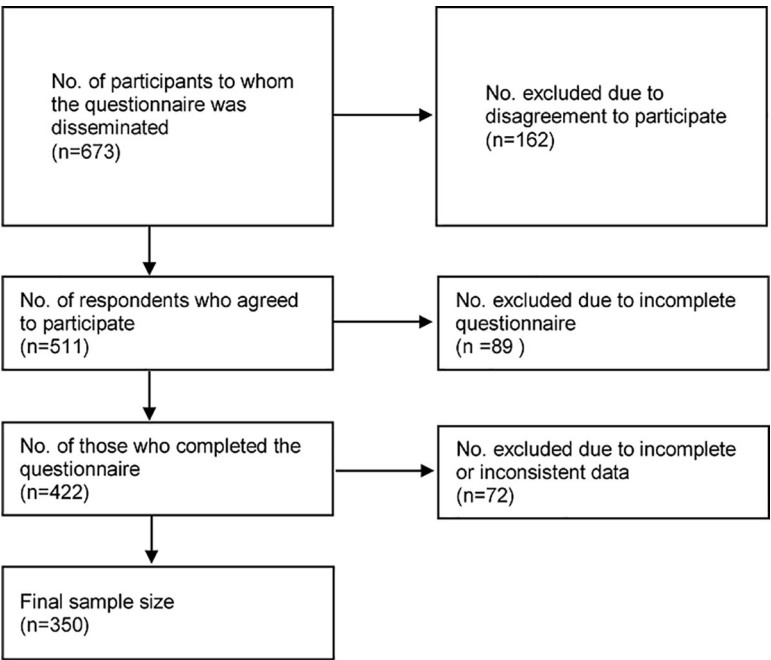

**Fig 1. Flow chart of the study population to validate the 5C scale.**

95% confidence interval (CI) to assess whether the 5C antecedents could predict the intention to get vaccine. We included in the model " intention to get the COVID-19 vaccine" as the dependent variable and the mean scores of the 5C sub-scales with the baseline criteria of the study participants as independent variables. P-value < 0.05 was considered statistically significant.

**Reliability and item analysis.**   Cronbach's alphas were calculated for the sub-scales of the questionnaire to assess its internal consistency. As a rule of thumb, a Cronbach's alpha of 0.70 to 0.80 is considered respectable for a scale for research use and an alpha more than 0.80 is considered very good [35].

**Construct validity.**   It represents the "extent to which an instrument assesses a construct of concern, and is associated with evidence that measures other constructs in that domain and measures specific real-world criteria" [28]. It is determined using content, criterion-related validity, and structural or factorial validity [36].

*Criterion-related validity.* concurrent, convergent, and discriminant (divergent) validity were used as indicators of criterion-related validity. Concurrent validity was assessed by determining whether the 5C antecedents predict the intention to get COVID-19 vaccine through multiple logistic regression analysis. Convergent validity was assessed by analyzing inter-item and item-to-mean score of the sub-scale correlation. Discriminant validity was assessed by calculating factor correlation matrix of the five subscales [37].

*Factorial analysis validity.* We analyzed data collected from 350 participants. Factor analysis was performed in two steps: exploratory and confirmatory factor analysis (EFA and CFA). We randomly divided the participants into two groups; 150 participants for EFA and 200 participants for CFA.

**Exploratory factor analysis.**   The EFA aimed at identifying the major factor structures for the set of 15 items and to determine the number of latent factors, without making assumptions about the factor relationships [38]. Kaiser-Meyer-Olkin (KMO) sampling adequacy measure and Bartlett's sphericity test were performed before EFA. The KMO statistics range from 0 to

1, with values closer to 1 denoting greater adequacy of the factor analysis (KMO ≥ 0.6 low adequacy, KMO ≥ 0.7 medium adequacy, KMO ≥ 0.8 high adequacy, KMO ≥ 0.9 very high adequacy) and P value of Bartlett's test is < 0.05, then factorial analysis can be used [39]. The number of factors extracted is based on Eigenvalues (>1), scree plot, parallel analysis, and interpretability of the factors [40].

To determine the type of rotation, we first ran EFA using the principal component analysis with an oblique direct Oblimin rotation to calculate the inter-factor correlation. Discriminant validity was assessed if inter-factors correlation based on the factor correlation matrix is less than 0.7 [41].

The final EFA was done using the principal component analysis with the orthogonal Varimax rotation. A factor loading cut-off value of 0.50 was chosen to decide which items were highly associated with a given factor [40]. In interpreting the output, we defined that each factor should have at least 3 items with high factor loadings of 0.5 and higher on the primary factor and minimal cross-loadings on any of the other factors (a < 0.35) to reduce the overlap between the sub-scales [40, 42].

**Confirmatory factor analysis.** The CFA that was performed based on the selected 200 participants aimed to measure how well the factor structure, identified in the EFA, fits the observed data. Specifically, we assessed the convergent and discriminant validity of the constructs and model fit measures using the SEM technique [43]. We used the root mean square error of approximation (RMSEA <0.08), comparative fit index (CFI >0.9), Tucker Lewis index (TLI>0.9), standardized root means square residual (SRMR ≤0.08), normal fit index (NFI>0.9), goodness of fit (GFI>0.9) as model fit indicators, and $\chi^2$/df <3 [44]. Convergent validity was determined if the average variance extracted (AVE) values of the different factors were above 0.5. Discriminant validity was confirmed if the square root of AVE is higher than the inter-correlation between the factors [45]. Moreover, we assessed the construct reliability of each latent factor and reliability ≥0.7 indicates good reliability [45]. We used statistical package of social science SPSS (version 25, Chicago, USA) and SPSS AMOS 26 to run all the analyses.

### Ethical considerations

The study was approved by the Ethics Committee of the Faculty of Medicine- Alexandria University, Egypt (IRB No:00012098) following the International Ethical Guidelines for Epidemiological studies [46].

## Result

### Characteristics of the study participants

Table 1 shows the baseline characteristics of the study population. Age ranged between 18 to 73 years; mean age of 34 ± 12 years. More than half were females (57.14%), 37.43% were living in Egypt, and 36.86% were living in UAE. As regards the nationality; 39.4% were Egyptians, 16.6% were Emirati, 2.9% were Moroccan, and 2.3% were Sudanese. One-third were healthcare workers and more than one-half (51.14%) were university graduates. Only 16.29% reported a previous history of COVID-19 infection, 38.57% gave a family history of death due to the infection, 79.42% reported knowing about the several types of vaccines, and 42.8% mentioned that they have the intention to get COVID-19 vaccine.

### Questionnaire validation

We ran univariate item analysis using collected data from 150 participants. All items means ranged from a minimum of 2.17 to a maximum of 6.14, and SD ranged from 1.25 to 1.94. Table 2 shows the descriptive statistics of the different items of the questionnaire (Table 2).

**Table 1. Baseline characteristics of study population.**

| Baseline characteristics | Frequency (%) |
|---|---|
| | (N = 350) |
| **Age** | |
| 18–30 | 104(29.71) |
| 31–45 | 149(42.57) |
| 46–60 | 71(20.57) |
| >60 | 25(7.14) |
| **Mean± SD age in years** | 34 ± 12 |
| **Sex** | |
| Male | 150(42.86) |
| Female | 200(57.14) |
| **Country** | |
| Egypt | 131(37.43) |
| Libya | 34(9.71) |
| United Arab of Emirates | 129(36.86) |
| Saudi Arabia | 56(16.00) |
| **Nationality** | |
| Egyptian | 138 (39.4) |
| Libyan | 34(9.7) |
| Lebanese | 24(6.9) |
| Syrian | 29(8.3) |
| Emirati | 58(16.6) |
| Saudi Arabian | 33(9.4) |
| Moroccan | 10(2.9) |
| Sudanese | 8(2.3) |
| Jordanian | 11(3.1) |
| Others | 5(1.4) |
| **Education** | |
| Secondary | 48(13.71) |
| Vocational education | 18(5.14) |
| University graduate | 179(51.14) |
| Post-graduate | 99(28.29) |
| Others | 6 (1.71) |
| **Chronic diseases** | |
| Yes | 75(21.4) |
| No | 275(78.57) |
| **Health care workers** | |
| Yes | 105(30.00) |
| No | 245(70.00) |
| **Did you get COVID-19 infection** | |
| Yes | 57(16.29) |
| No | 225 (64.29) |
| I do not know | 68(19.42) |
| **If there any of your relative died due to COVID-19 infection** | |
| Yes | 135(38.57) |
| No | 215(61.43) |
| **Do you know that there is many types of COVID-19 vaccine** | |
| Yes | 278 (79.42) |

(*Continued*)

**Table 1.** (Continued)

| Baseline characteristics | Frequency (%) |
|---|---|
| | **(N = 350)** |
| No | 72(20.58) |
| Do you have the intention to get COVID-19 vaccine? (n = 339)$ | |
| **Yes** | 145(42.8) |
| **No** | 194(57.2) |

*others (1 from Tunisia, 1 from Algeria, 1 from Mauritania, 1 from Bahrain)

$ There are 11 participants that did not answer this question

**Reliability analysis.** All sub-scales had a satisfactory internal consistency. Both "Confidence" and "Collective responsibility" sub-scales have Cronbach's alpha of 0.829." "Constraints" sub-scale had the lowest Cronbach's alpha (0.701) (Table 2).

**Concurrent validity.** Table 3 showed that intention to get COVID-19 vaccine was predicted by age as younger people (aged less than 40 years) were 85% more intended to get COVID-19 vaccine compared to older participants (OR: 1.85, 95%CI:1.07–3.21). The 5 C sub-scales were significantly predicting the intention to get COVID-19 vaccine as follows; confidence (OR: 1.15, 95%CI:1.07–1.69), complacency (OR: 0.91, 95%CI: 0.86–0.98), constraints

**Table 2. Descriptive statistics, reliability, and convergent validity of the Arabic version of the 5C scale.**

| Variable | Mean ± SD | Item-mean score correlation |
|---|---|---|
| **Confidence** | | |
| **Q1** | 4.65±1.73 | 0.91($P<0.001$) |
| **Q2** | 4.93±1.57 | 0.87($P<0.001$) |
| **Q3** | 5.15±1.92 | 0.82($P<0.001$) |
| **Cronbach's alpha** | | **0.829** |
| **Complacency** | | |
| **Q4** | 2.17±1.79 | 0.81($P<0.001$) |
| **Q5** | 3.76±1.86 | 0.79($P<0.001$) |
| **Q6** | 3.07±1.94 | 0.79($P<0.001$) |
| **Cronbach's alpha** | | 0.712 |
| **Constraints** | | |
| **Q7** | 2.81±1.77 | 0.70($P<0.001$) |
| **Q8** | 3.12±1.79 | 0.82($P<0.001$) |
| **Q9** | 2.75±1.85 | 0.79($P<0.001$) |
| **Cronbach's alpha** | | 0.701 |
| **Calculation** | | |
| **Q10** | 5.51±1.67 | 0.84($P<0.001$) |
| **Q11** | 5.76±1.39 | 0.86($P<0.001$) |
| **Q12** | 6.14±1.36 | 0.80($P<0.001$) |
| **Cronbach's alpha** | | 0.773 |
| **Collective responsibility** | | |
| **Q13** | 5.85±1.25 | 0.80($P<0.001$) |
| **Q14** | 5.60±1.73 | 0.91($P<0.001$) |
| **Q15** | 6.03±1.37 | 0.89($P<0.001$) |
| **Cronbach's alpha** | | 0.829 |

**Table 3. Predictors of intention to get COVID-19 vaccine among the study participants.**

| Variables | OR (95% CI) | p-value |
|---|---|---|
| **Age categories** | | |
| <40 years | 1.82(1.07–3.21) | 0.02 |
| > = 40 years* | 1 | - - - |
| **Sex** | | |
| Male | 1.26(0.78–2.06) | 0.34 |
| Female* | 1 | - - - |
| **Country** | | |
| Egypt | 0.43(0.23–0.79) | 0.06 |
| Libya | 0.54(0.22–1.33) | 0.18 |
| United Arab of Emirates | 0.52(0.24–1.10) | 0.07 |
| Saudi Arabia* | 1 | |
| **Education** | | |
| Before university | 1.43(0.69–2.92) | 0.33 |
| University graduates | 0.93(0.52–1.67) | 0.80 |
| Post-university graduates* | 1 | - - - |
| **Chronic diseases** | | |
| Yes | 0.72(0.38–1.45) | 0.30 |
| No* | 1 | - - - |
| **Health care workers** | 1 | |
| Yes | 1.07(0.61–1.88) | 0.81 |
| No* | 1 | - - - |
| **Did you get COVID-19 infection** | | |
| Yes | 1.34(0.87–2.06) | 0.18 |
| No and don't know* | 1 | - - - |
| **If there any of your relative died due to COVID-19 infection** | | |
| Yes | 0.69(0.54–1.58) | 0.88 |
| No* | 1 | - - - |
| **Do you know that there is many types of COVID-19 vaccine** | | |
| Yes | 0.75(0.38–1.49) | 0.41 |
| No* | 1 | - - - |
| **5C scale** | | |
| Confidence | 1.15(1.07–1.69) | 0.02 |
| Complacency | 0.91(0.86–0.98) | 0.01 |
| Constraints | 0.88(0.82–0.94) | <0.001 |
| Calculation | 1.08(1.01–1.16) | 0.04 |
| Collective responsibility | 1.07(1.03–1.14) | 0.03 |

*reference group

(OR: 0.88, 95%CI: 0.82–0.94), calculation (OR: 1.08, 95%CI:1.01–1.16), and collective responsibility (OR: 1.07, 95%CI: 1.03–1.14).

**Convergent validity.** Inter-item correlation for each sub-scale was highly significant (P<0.001) (S3 Table). In addition, item-mean score of the sub-scale correlation was significant. (Table 2).

**Exploratory factorial analysis.** Before conducting the EFA, we assessed the sampling adequacy and sphericity assumptions. KMO measure of sampling adequacy was 0.80, which is above the recommended value of 0.60, and Bartlett's test of sphericity was found to be highly significant ($P < 0.001$). Moreover, all the communalities demonstrated to be 0.5 or more.

**Table 4. Factor correlation matrix of the Arabic version of the 5C scale.**

| Factor | Confidence | Complacency | Constraints | Calculation | Collective responsibility |
|---|---|---|---|---|---|
| Confidence | 1.000 | | | | |
| Complacency | -0.208 | 1.000 | | | |
| Constraints | 0.300 | -0.276 | 1.000 | | |
| Calculation | -0.077 | -0.074 | 0.226 | 1.000 | |
| Collective responsibility | -0.174 | 0.241 | -0.248 | 0.033 | 1.000 |

Using these previously mentioned indicators, we conducted an EFA; at first, we ran the analysis in the form of principal component analysis with an oblique direct Oblimin rotation to assess the factor correlation matrix and check the discriminant validity. There were both negative and positive correlations among the five factors. The largest negative correlation was between Complacency and Constraints (-0.276), while the smallest negative correlation was between Complacency and Calculation (-0.074). The largest positive correlation was between Confidence and Constraints (0.300), while the lowest positive correlation was between Calculation and Collective responsibility (0.033). There were no correlation coefficients larger than 0.7; hence, the factors derived from EFA revealed adequate **discriminant validity** (See details in Table 4).

The final analysis took the form of the principal component analysis with Varimax rotation. The initial Eigenvalues showed that all 15 items of the questionnaire explained 72.8% of the variance in 5 factors. Table 5 shows the factor loadings for all items of the questionnaire. For "**Confidence sub-scale**," the items were loaded on one factor with loading ranges from 0.782 to 0.868. For **the "Complacency sub-scale**," all items were loaded on one factor with factor loading ranges from 0.736 to 0.793. For "**Constraints sub-scale**," items loaded on one factor, with loadings from 0.606 to 0.861. For "**Calculation sub-scale**," the items loaded on one factor, with loadings between 0.726 to 0.863. Lastly, for "**Collective responsibility**," all items loaded on one factor with factor ladings ranges between 0.478 to 0.808.

**Confirmatory factor analysis.** To determine whether EFA proposed five-factor model with the 15-item questionnaire can be used as a valid tool towards assessment of the

**Table 5. Factor loadings of the Arabic version of 5C scale.**

| Items | Factor | | | | | Communalities |
| | Confidence | Complacency | Constraints | Calculation | Collective responsibility | |
|---|---|---|---|---|---|---|
| Q1 | **0.875** | -0.158 | -0.106 | -0.125 | 0.182 | 0.851 |
| Q2 | **0.833** | -0.245 | -0.016 | -0.029 | 0.193 | 0.792 |
| Q3 | **0.758** | 0.216 | -0.091 | 0.027 | 0.252 | 0.693 |
| Q4 | -0.270 | **0.772** | 0.255 | -0.155 | 0.016 | 0.758 |
| Q5 | 0.074 | **0.774** | 0.074 | 0.115 | -0.214 | 0.669 |
| Q6 | -0.060 | **0.745** | 0.042 | 0.007 | -0.143 | 0.581 |
| Q7 | 0.124 | 0.026 | **0.854** | -0.117 | 0.002 | 0.76 |
| Q8 | -0.323 | 0.258 | **0.657** | 0.068 | -0.245 | 0.667 |
| Q9 | -0.394 | 0.215 | **0.571** | 0.049 | -0.261 | 0.598 |
| Q10 | -0.087 | 0.000 | 0.074 | **0.834** | -0.010 | 0.708 |
| Q11 | 0.073 | -0.013 | -0.020 | **0.868** | 0.148 | 0.78 |
| Q12 | -0.112 | 0.030 | -0.160 | **0.734** | 0.319 | 0.68 |
| Q13 | 0.320 | -0.316 | -0.265 | 0.079 | **0.594** | 0.632 |
| Q14 | 0.301 | -0.125 | -0.091 | 0.141 | **0.817** | 0.802 |
| Q15 | 0.220 | -0.156 | -0.088 | 0.319 | **0.825** | 0.862 |

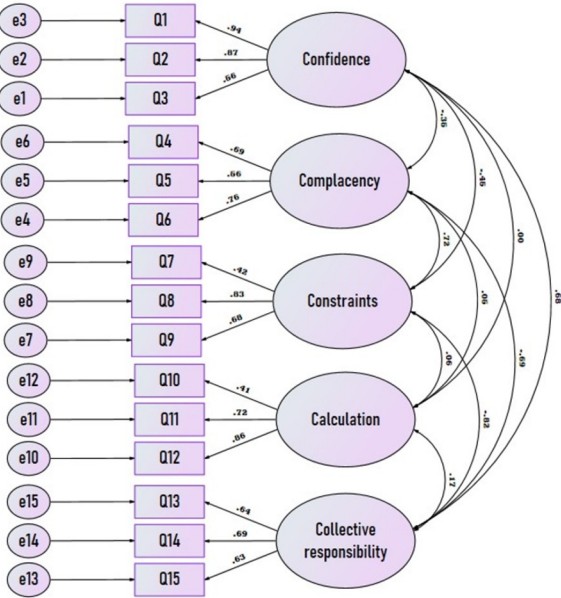

**Fig 2. Confirmatory factor analysis of the 15 questions related to the 5 domains of 5C scale of vaccine antecedent.**

psychological antecedents of COVID-19 vaccines among the Arab population, we conducted a CFA using a different sample of 200 participants.

We ran the CFA on the 15 items. We described the results of the CFA final model with the SEM shown in Fig 2. All the loadings were from 0.41 to 0.94. The construct reliability of the five factors in the CFA final model were above 0.7. For convergent validity, the average variance extracted (AVE) values of **confidence**, **complacency** and **calculations** factors were above 0.5. Although the AVE value of **constraints and collective responsibility** factors were less than 0.5, the factors specific items loadings were acceptable for convergent validity since there were no items with loading below 0.4. The correlation between the five latent variables was less than squared root of AVE, hence no problem with discriminant validity.

An overview of goodness-of-fit measures for the final model is presented in Table 6. The results demonstrate good model-data-fit, i.e., RMSEA <0.08, GFI, NFI, CFI, and TLI >0.9, and SRMR<0.08. Hence, the 15-item questionnaire has good psychometric properties and model fit to observed data.

## Discussion

Public hesitancy to COVID-19 vaccines can hamper the international efforts to mitigate COVID-19 infection. Vaccine hesitancy and refusal are significant concerns globally, prompting the WHO to declare this uncertainty among the top 10 health threats in 2019 [47]. There is an increased need to assess the COVID-19 VH and the challenges for facing it. Existing literature showed that religious reasons, personal beliefs, risk perceptions, and safety concerns due to wide-spread myths are the main determinants of VH. Those with higher VH are more likely to have beliefs, a lack of trust in those responsible for health and lower levels of compliance with public health advice for COVID-19 [48–50].

To our knowledge, an Arabic validated instrument that may evaluate the COVID-19 VH in the Arab world does not exist. In this paper we validated the 5C psychological antecedents'

**Table 6. Results of the confirmatory factor analysis of 5C scale (15 items).** Convergent validity, discriminant validity, and reliability assessment of CFA final model with five latent factors and model fit indices.

| Factor | CR[a] | AVE[b] | Correlations among latent variables | | | | |
|---|---|---|---|---|---|---|---|
| | | | Confidence | Complacency | Constraints | Calculations | Collective responsibility |
| **Confidence** | 0.843 | 0.651 | **0.807** | | | | |
| **Complacency** | 0.712 | 0.501 | -0.346 | **0.675** | | | |
| **Constraints** | 0.690 | 0.442 | -0.448 | 0.623 | **0.665** | | |
| **Calculations** | 0.719 | 0.510 | 0.001 | 0.065 | 0.065 | **0.692** | |
| **Collective responsibility** | 0.689 | 0.426 | 0.602 | -0.69 | -0.816 | 0.174 | **0.652** |
| **Model Fit indices** | RMSEA[c] | | GFI[d] | CFI[e] | TLI[f] | SRMR[g] | NFI[h] |
| | 0.060 | | 0.924 | 0.957 | 0.937 | 0.076 | 0.906 |

a) construct reliability

b) average variance explained

c) root mean square error of approximation

d) goodness of fit index

e) comparative fit index

f) Tucker-Lewis Index

g) standardized root mean square residual

h) normal fit index.

questionnaire among a selected Arabic speaking population from four randomly selected Egypt, Libya, UAE, and Saudi Arabia.

Using a validated tool allows for better transparency and improves opportunities to decrease researchers' bias. Differences between regions, populations, and cultures require reliability and validity assessment of measurement instruments [51]. Although different dialects are used, formal Arabic is the official language regardless the geographical location. For that, we used the formal Arabic language to translate and validate the 5C questionnaire among Arab populations.

Based on our findings, the psychometric results of the Arabic version of 5C scale were close to the values of the corresponding items in the original German questionnaire [29]. Lower value of Cronbach's alpha was obtained from constraints sub-scale (0.70) compared to the original questionnaire (0.85). On the other hand, the Arabic version of the questionnaire showed a higher Cronbach's alpha (0.83) for the collective responsibility sub-scale compared to the original questionnaire (0.71). This may be explained by the different context in which we tested the 5C scale. While the original questionnaire was tested before the era of COVID-19 pandemic, our questionnaire was peculiarly validated for COVID-19 vaccines. The debates about the different vaccines efficacy and safety affect the Arab population acceptance. In addition, the vaccines are not widely administered in all countries due to different polices regarding the eligibility and stock availability. The construct validity showed that five factors structure was extracted, which is similar to the original copy of the 5C questionnaire [29].

Among our study population, 42.8% showed the intention to get COVID-19 vaccines. This was close to what has been reported from other Arab studies [20, 22] and lower than that from Saudi Arabia [21]. Younger age, stronger confidence and collective responsibility, higher constraints, and weaker complacency were associated with stronger intention to get COVID-19 vaccine. Young participants were more likely to accept COVID-19 vaccines in the current study, which is similar to what has been reported from other studies [20, 24, 52]. In contrast, other previous studies showed higher acceptance among older age [13, 21, 53]. Young people are more frustrated with social restrictions and curfews associated with the COVID-19 crisis

and may show more willingness to be vaccinated. At the same time, younger people may be more accustomed and trusting of science and technology in contrast with their older counterparts. Moreover, school suspension may negatively affect the academic performance of school-aged and university participants. Therefore, they are more impatient to bring an end to the situation and thus more accepting of vaccination [54–56].

The psychological antecedents of the 5C scale were able to predict the intention to get COVID-19 vaccines as shown in other studies [57–59]. The speed at which vaccines have been developed, which reflects the unprecedented amount of funding from governments and non-profit groups, has raised concerns that the trials were rushed and regulatory standards relaxed. Also, there are no previously approved mRNA vaccines, which has also sparked hesitancy given the novelty of the approach. Lastly, conspiracy theories about COVID-19 vaccines are being widely circulated on unregulated social media platforms, sometimes by highly organized anti-vaccination group. Vaccine acceptance could be strengthened by increasing the knowledge and awareness, community engagement, and more manufacturers obtain authorization from stringent regulatory authorities or WHO and by these bodies clearly communicating to the public the rationale behind their decisions [60].

A strong public health implication of this study is that the Arabic validated 5C scale will help in understanding people's readiness, confidence, perceptions, psychological and cultural antecedents toward the COVID-19 vaccination. This will guide the local public health authorities to design targeted vaccine interventional programs and allow the comparison between different countries regarding the vaccination coverage achievements. Understanding the factors and determinants for COVID-19 vaccine acceptance will also improve the efficiency of these roll out campaigns.

## Strengths and limitations

The strength of our study lies in being the first one to validate the 5C tool to be used in the assessment of COVID-19 vaccine hesitancy among the Arab population, along with including study population from four Arab countries with different Arab nationalities. However, we acknowledge that there are few limitations. The first one is that the study was conducted as a web-based survey that may introduce selection or no-response bias. However, it was in alignment with the research objectives as it guided the large-scale survey administration during a period when restrictions were enforced. This technique ensured the safety of both interviewers and interviewees. Second, the study was a cross sectional one that does not allow for assessment of the changes in the COVID-19 vaccine acceptance over time after the widespread campaigns to motivate population to get COVID-19 vaccine. However, we thought that it would not affect the stability of responses as the Arabic version of 5C questionnaire showed high reliability. Third, we did not test the validity of the 5C questionnaire among Arab population living in Western countries, however, this will be considered in the other part of our project to assess the vaccine hesitancy among Arab population living inside and outside the Arab region. Finally, we used non- random sampling technique (convenience sampling method) for including the study population, however, this method was the most appropriate due to extended lockdown and poor access to the community members.

## Conclusion

This study provides evidence on the adequate validity and reliability of the Arabic version of the 5C scale to assess the psychological antecedents to COVID-19 vaccine.

## Supporting information

**S1 Table. The English version of the 5C questionnaire.**
(PDF)

**S2 Table. The Arabic validated 5C questionnaire.**
(PDF)

**S3 Table. Inter-item correlations of the Arabic version of the 5C scale.**
(PDF)

## Acknowledgments

The authors would like to acknowledge and thank all study participants.

## Author Contributions

**Conceptualization:** Samar Abd ElHafeez, Ramy Mohamed Ghazy.

**Data curation:** Haider M. El Saeh, Ramy Mohamed Ghazy.

**Formal analysis:** Samar Abd ElHafeez, Ramy Mohamed Ghazy.

**Investigation:** Iffat Elbarazi.

**Methodology:** Samar Abd ElHafeez.

**Project administration:** Ramy Mohamed Ghazy.

**Resources:** Samar Abd ElHafeez, Iffat Elbarazi, Ramy Shaaban, Rony ElMakhzangy, Maged Ossama Aly, Mohamed Yacoub, Haider M. El Saeh, Nashwa Eltaweel, Sulafa T. Alqutub.

**Software:** Ramy Shaaban.

**Supervision:** Samar Abd ElHafeez, Ramy Mohamed Ghazy.

**Validation:** Samar Abd ElHafeez, Sulafa T. Alqutub, Ramy Mohamed Ghazy.

**Writing – original draft:** Samar Abd ElHafeez, Iffat Elbarazi, Ramy Mohamed Ghazy.

**Writing – review & editing:** Samar Abd ElHafeez, Iffat Elbarazi, Ramy Shaaban, Rony ElMakhzangy, Maged Ossama Aly, Amr Alnagar, Mohamed Yacoub, Haider M. El Saeh, Nashwa Eltaweel, Sulafa T. Alqutub, Ramy Mohamed Ghazy.

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
