## [Decision Letter · Decision Letter 0]

7 Apr 2021

PONE-D-21-03821

Arabic validation and cross-cultural adaptation of the 5C scale for assessment of COVID-19 vaccines psychological antecedents

PLOS ONE

Dear Dr. Elbarazi,

Thank you for submitting your manuscript to PLOS ONE. After careful consideration, we feel that it has merit but does not fully meet PLOS ONE’s publication criteria as it currently stands. Therefore, we invite you to submit a revised version of the manuscript that addresses the points raised during the review process.

The reviewers were unanimous on the need to further improve the methodological clarity of the paper and highlighted multiple weaknesses in the introduction and discussion sections. I  agreed with the concerns of the three reviewers and would encourage you to submit a revision addressing these concerns as much as possible.

We look forward to receiving your revised manuscript.

Kind regards,

Adewale L. Oyeyemi, Ph.D

Academic Editor

PLOS ONE

Journal Requirements:

1. Please ensure that your manuscript meets PLOS ONE's style requirements, including those for file naming. The PLOS ONE style templates can be found athttps://journals.plos.org/plosone/s/file?id=wjVg/PLOSOne_formatting_sample_main_body.pdf and https://journals.plos.org/plosone/s/file?id=ba62/PLOSOne_formatting_sample_title_authors_affiliations.pdf

Additional Editor Comments (if provided):

Reviewers' comments:

Reviewer's Responses to Questions

**Comments to the Author**

1. Is the manuscript technically sound, and do the data support the conclusions?

Reviewer #1: Yes

Reviewer #2: Yes

Reviewer #3: Yes

2. Has the statistical analysis been performed appropriately and rigorously? 

Reviewer #1: Yes

Reviewer #2: Yes

Reviewer #3: I Don't Know

3. Have the authors made all data underlying the findings in their manuscript fully available?

Reviewer #1: Yes

Reviewer #2: Yes

Reviewer #3: Yes

4. Is the manuscript presented in an intelligible fashion and written in standard English?

Reviewer #1: Yes

Reviewer #2: Yes

Reviewer #3: Yes

5. Review Comments to the Author

Reviewer #1: Introduction

* Authors mention in the introduction the use of different instruments that measure vaccine acceptance. Why did they use this particular instrument 5C? How it compares with others?

* Need some more information about the history of the development of this instrument, where has it been previously used, among which populations and at what behavioral settings, were there any psychometric properties established, has the instrument been applied successfully in other settings? Are there any limitations regarding using this instrument?

* It is stated that certain countries like Kuwait and Jordan have lower acceptance rate of the vaccine. Why is that? How do the authors explain this hesitancy within the population of these two Arab countries?

* No need to include the word “respectively” since the percentages are shown next to each country.

* Why is this study significant from a measurement perspective but most importantly from a public health perspective?

* What is the research question that this study is trying to answer?

Methods:

* The items of each scale need to be included in a narrative format and not in bullet format. There should be an introductory sentence like: The following are the items of the scale.

* There is a lot of quoting, avoid quotations as much as possible and simply paraphrase.

* In the methods sections, the other demographic variables are not included. These should be also included in the description of the variables besides the cognitive variables of the 5C

* It is not clear why the authors did not calculate the score of a variable. Can they give an example? Also, why didn’t the authors create a dependent variable like the intention to get the vaccine once available ao that they can test the predictability of these variables since they were planning to collect the data anyways?

* Regarding the cognitive interviews, can the authors give examples of the questions they asked? How much time each cognitive interview took place? How did they analyze the results? What is the demographic picture of the people participating in the cognitive interviews? Were the interviews in a sense the pretesting of the survey? This is not clear.

* I am not sure if I understand whether the authors claim to have used a random sample or not. They state that the countries selected were random. It is hard to believe that because to do such a multi-country study one needs to have established collaborative relationships with researchers from that country. In addition, the survey was distributed via social media, whatsapp etc. I guess, one person was forwarding the email to another person, correct? In other words, this was convenience sampling and not random. A random sample would have been if a huge list of potential participants was given to the researchers and they chose randomly 600 of those and sent them the survey via email etc. How were the participants recruited? It is not clear.

* Under data management and psychometric analysis the authors write that “ qualitative data were presented with percent and frequency”. Qualitative data refers to text and comments. Did the authors mean something else?

* Was the survey written in classical or formative Arabic language? Were there any difficulties in understanding some of the words even in classical Arabic given the different dialects?

Results:

* The results seem well written however, one can see from the demographics that the study sample is skewed towards the higher educated populations since almost 80% of the study population were at least University graduates. Another observation is that authors do not mention the ethnic group of the participants, just the country. Were all participants Arabs? This was not clear.

* Another observation is that more than one third of the participants had a relative who died from COVID. This could have influenced the motivation to accept the vaccine. I would suggest that one of the exclusion criteria in the recruitment process was having a relative who died from Covid because this subpopulation might have different motivations toward getting the vaccine compared to the general public because of witnessing a loved one dying from COVID. Were any exclusion criteria established?

Discussion

* The authors claim that the results of the study are representative of the Arab region. This is true up to an extent, countries like Lebanon, Jordan and Syria, the Levant region are not included. Are those represented as well?

* The discussion was very shallow and it is focused mostly on the psychological interpretations of the results. The authors used vague terms like “ the individual” or the “people”. Well these people are Arab people. I suggest that the author interpret the result through the lenses of Arab culture and Islamic religion. For instance it is well known, that Arab societies are collectivist societies, how does that relate to the results? What about the trust in God or fate, that if someone gets sick it was meant to happen? Or that disease is perhaps punishment from God? How can public health professionals who plan interventions to promote vaccine acceptance in these countries can use the results of this study?

* How do the authors interpret the results based on the fact that almost a third of the sample had someone who had died from COVID? Do they think that might have influenced the way the participants answered the questions or not?

* What are the study limitations?

Reviewer #2: In general, it is a very interesting topic specially that validations scales in Arabic are very needed studies. Researches have done good job in the statistical analysis of the study and explained their methodology very well. Few suggestions can be provided to make the paper sound even better:

Introduction:

1. The flow of the introduction can be improved paragraph1 is about the outbreak of COVID-19. Paragraph2 facing COVID-19 globally. Paragraphs 3,4 and 5 can be combined about the COVID-19 vaccine. Other paragraphs can be used to introduce: acceptance of the vaccine globally, if there are different tools used to understand vaccine psychological impact, if there any theoretical and conceptual framework used to develop the current tool, and finally stating the aim of the study.

Method:

2. The method section is too long! Is there a way to shorten it? Keep important information in the method section and extra information in the supporting document if this is possible.

3. For study tool for example, keep the sub-scales and their definitions. Categories of the subscales can be moved in the supporting document.

4. Although the mode of the survey was mentioned, it is not clear to me how the study targeted the population of the four countries Egypt, Libya, UAE and KSA? The availability of the survey online means that the survey was open to anyone from any country? What measures or strategies researchers did to ensure targeting the specified countries mentioned? This need to be explained to remove ambiguity.

5. Score interpretation: can you further clarified, and refined in a simpler way for the readers to understand.

6. Translation and adaptation: how many total researchers/co-authors/ translators worked for this section? This section can be refined by stating the total number of individuals who worked on this section and then each stage of the translation should state the number of individuals who worked at that stage.

7. Translation and adaptation: are the same translators who translated the tool into Arabic assessed the translation or separate individuals? Just to avoid bias

8. Cognitive interviews: you mean piloting right? What about adding the word “pilot” since people are more familiar with the word pilot compared to cognitive interview? Or is it because it is a psychological study the term cognitive interview is preferred compared to the word piloting?

9. Do you think this paragraph: Based on the sample size recommendations of having 10 participants respond to each item for validating a questionnaire (ratio 10:1), we needed 150 participants [25]. 8 Moreover, a priori sample size calculation for Structural Equation Modelling (SEM) technique to perform confirmatory factor analysis (CFA) showed that a minimum sample of 200 is required to run CFA [26]. For that, the minimum required sample size for our analysis was 350 participants. Adult (18 years and above) Arabic speaking population is included in the study, should be labeled as a sample size calculation or study sample size?

10. Do you think this paragraph: The final Arabic copy of 5C scale was uploaded on Qualtrics and disseminated online via different social media platforms (Facebook, WhatsApp, emails, and Twitter) to 673 participants? The sample was recruited from four randomly selected Arabic countries (Egypt, Saudi Arabia. Libya, and United Arab of Emirates (UAE)). A total of 511 responded to the questionnaire, 89 participants chose not to complete the questionnaire. The response rate was 62.70% (422/673). Of the 422 who completed the questionnaire, we excluded 72 responses from the final analysis due to incomplete or inconsistent data. The final sample size included in our analysis was 350 participants, could be labeled as data source?

11. What about adding: time frame of the study? When did the data collection start and ended? For how long the survey was open to participants?

12. Do you think converting the paragraph above to “figure as a flow chart for participant recruitment”, will make it easier for the reader?

13. What about clarifying the design of the study and stating that it is a quantitative and a cross sectional study?

Discussion:

14. Discussion should focus on interpreting the main results of the study no repeating the results

15. Comparing the results of the Arab words to studies conducted elsewhere (missing?)

16. Strengths and limitation of the study (missing?)

Conclusion:

17. Why there is no separate conclusion to this study?

18. Conclusion should summarize main findings, any policy implications of the study, any recommendations, any future research to answer or understand questions raised by this study

Reviewer #3: This manuscript is well written. However, there are some comments that need clarifications.

Introduction:

Page 4 second paragraph: Authors need to differentiate between the acceptability and the availability of the vaccine in these listed countries. As these countries varies in their economic status and the availability/ affordability of the vaccine.

Methodology:

the study design should be clearly mentioned, and the authors should make is clear that they used a mixed approach (both qualitative and quantitative). Also, the reasons behind using the mixed approach (quantitative and the qualitative).

Its not clear how the participants were recruited from four selected Arabic countries? how did they select the four countries randomly? (Egypt, Saudi Arabia. Libya, and United Arab of Emirates (UAE). The authors mentioned that the four Arab countries were randomly selected. it’s not clear how they selected these countries in a random basis? and why four and not more or less?

By reading the manuscript, I thought that the sampling methodology is non-proportional convenient sampling. Those included in the study are participants who are available and volunteered to participate.

Page 8 at the end of paragraph 1 " we excluded 72 responses from the final analysis due to incomplete or inconsistent data". Are these questionnaires distributed over selected countries? what is the percentage from each country?

End of page 8, beginning of page 9, its not clear how the researchers measure the construct validity and how they randomly divide the sample into 2 groups one with 150 and the other with 200?

The translated Arabic version of the questionnaire should be provided with the manuscript.

Results:

Because there is high discrepancy between the lowest and the highest age that affected the mean, its better to categorize the age and then run the analysis. Also, about one third of the participants are from Egypt whose Arabic accent and words meaning to some extent is different than other Arab countries. Also, none of the following countries are included in the sample (Jordan, Lebanon, Syria, and Palestine) who have seminaries in Arabic while they are different than other Arab countries, this might create bias in sample selection and the results.

Discussion

Page 11: I disagree with the authors statement “Therefore, chosen countries in this study are good representative of the Arab region". They missed to include one representative country from: (Jordan, Lebanon, Syria, and Palestine). Authors should mention that.

6. PLOS authors have the option to publish the peer review history of their article (what does this mean?). If published, this will include your full peer review and any attached files.

Reviewer #1: No

Reviewer #2: No

Reviewer #3: **Yes: **Haleama Al Sabbah

---

## [Author Response · Author response to Decision Letter 0]

26 May 2021

RESPONSE TO REVIEWERS

Dear Editor

We would like to thank you for the opportunity to revise our manuscript and to improve it as per the reviewers’ comments. Please find our responses below. We would like to thank all three reviewers for their comments and their recommendations.

Comments to the Authors

1. Is the manuscript technically sound, and do the data support the conclusions?

Reviewer #1: Yes

Reviewer #2: Yes

Reviewer #3: Yes

2. Has the statistical analysis been performed appropriately and rigorously?

Reviewer #1: Yes

Reviewer #2: Yes

Reviewer #3: I Don't Know

3. Have the authors made all data underlying the findings in their manuscript fully available?

Reviewer #1: Yes

Reviewer #2: Yes

Reviewer #3: Yes

4. Is the manuscript presented in an intelligible fashion and written in standard English?

Reviewer #1: Yes

Reviewer #2: Yes

Reviewer #3: Yes

5.Reviewers comments and authors’ responses

Reviewer #1:

 Introduction

* Authors mention in the introduction the use of different instruments that measure vaccine acceptance. Why did they use this particular instrument 5C? How it compares with others?

Response: Thank you for your comment. We expanded on this in the introduction [P: 5&6, L:147-155] as’’ This tool expands the scope of the measures and the theoretical and conceptual frameworks used to study VH and acceptance. All available tools depend on 3C model (confidence, complacency, constraints) to assess the VH. The 5C scale provides more in-depth understanding of the “individual mental representations, attitudinal and behavioural tendencies that are a result of the environment and context the respondent lives in”. It assesses five psychological determinants pertaining to the individual’s vaccination decision: confidence, complacency, constraints, calculation, and collective responsibility. As a limitation, 5C scale authors pointed out the difficulty of generalizing the predictive validity of the 5C tool if not tested in other countries and on other populations’’

*Need some more information about the history of the development of this instrument, where has it been previously used, among which populations and at what behavioral settings, were there any psychometric properties established, has the instrument been applied successfully in other settings? Are there any limitations regarding using this instrument?

Response: Thank you for your comment. We clarified this and added more information about the instrument in the introduction [P5:, L147-148]. Being a new instrument, to our knowledge it has not been tested yet in other populations. However, the instrument is currently being validated in English and German (a protocol for the validation was published in 2020). We compared the psychometric properties of the English and German versions against our Arabic copy in the discussion section [P: 15 &16, L: 424-450].

* It is stated that certain countries like Kuwait and Jordan have lower acceptance rate of the vaccine. Why is that? How do the authors explain this hesitancy within the population of these two Arab countries?

Response: Thank you for raising this point. We elaborated on this based on the authors comment in the introduction [P 5: L:138-140] . 

* No need to include the word “respectively” since the percentages are shown next to each country.

Response: Thank you for your note. We removed it

* Why is this study significant from a measurement perspective but most importantly from a public health perspective?

Response: We do appreciate your enquiry. We think that this study is important from a public health perspective since COVID-19 vaccines became available in most of the Arab countries and based on our observations as healthcare workers, we noticed that people are hesitant in accepting the vaccine. Several proposals have been submitted to assess the vaccine acceptance among the Arab population. Neither of them used a validated tool to assess the vaccine hesitancy which in turn could affect the internal validity of the different studies and won’t allow for comparing the results and impact the external validity. For that, we proposed this research question to validate a study tool into Arabic so as to be used for assessing the vaccine hesitancy among different Arabic countries [P:6, L:163-167]. This in turn will help in understanding people’s readiness, confidence, perceptions, psychological and cultural antecedents toward the COVID-19 vaccination that will guide the local public health authorities to design targeted vaccine interventional programs [P 16:, L:463-469].

* What is the research question that this study is trying to answer?

Response: we try to answer the following question : is the Arabic translated copy of 5C questionnaire a valid and reliable tool to assess the vaccine hesitancy among general population in the Arab countries? [P:6,L:163-167].

Methods:

* The items of each scale need to be included in a narrative format and not in bullet format. There should be an introductory sentence like: The following are the items of the scale.

Response: Thank you for your suggestion. Now, we modified it [P:7, L:176-193].

* There is a lot of quoting, avoid quotations as much as possible and simply paraphrase.

Response: We thank the reviewer for his comment, we have reduced the quotations in the manuscript.

* In the methods sections, the other demographic variables are not included. These should be also included in the description of the variables besides the cognitive variables of the 5C.

Response: We do agree with you and we add it to the method section [P:6&7, L:176-181].

* It is not clear why the authors did not calculate the score of a variable. Can they give an example? Also, why didn’t the authors create a dependent variable like the intention to get the vaccine once available as that they can test the predictability of these variables since they were planning to collect the data anyways?

Response: Thank you for pointing out this. We explained the way of score interpretation in the manuscript as following’’ Each of the 5 sub-scales (confidence, complacency, constraints, calculation, and collective responsibility), was assessed by 3 rating items on a 7-point scale (1 = strongly disagree; 7 = strongly agree). The mean scores of items under each sub-scale were computed, with higher score indicating stronger agreement of the corresponding sub-scale. Using the 5C scale does not lead to a total score providing a sample’s absolute state of hesitancy. It, rather, allows for a valid assessment of the different psychological antecedents’’ [P:7, L:194-202]. We also contacted the authors of the original 5C questionnaire to understand the way of interpreting the score and they advised that the scores for each sub-scale should be interpreted in the context where it will be used as cutoff points could differ based on the background culture and environment. Hence, we worked on determining the cutoff points for 5C scale in another manuscript (accepted for publication).

For the second point raised by the reviewer one related to intention to get the vaccine, we already had data on this variable as this manuscript is a part of a large study to assess the vaccine hesitancy among the Arab population. In this project, we worked on three main pillars; validate 5C questionnaire, determine the cutoff points for 5C scale, and then use the validated questionnaire to assess the psychological antecedents for COVID-19 vaccine among Arab population. We supposed that the variable’’ intention to get vaccine’’ suits more the other aims, that is why we did not add it here. Now, we presented data on intention to get the vaccine in table 1 and results [P:12, L:342-343].We also assessed the ability of the 5C sub-scales to predict the intention to get COVID-19 vaccine after adjustment of other variables (table 3) [P: 12, L: 352-358].

* Regarding the cognitive interviews, can the authors give examples of the questions they asked? How much time each cognitive interview took place? How did they analyze the results? What is the demographic picture of the people participating in the cognitive interviews? Were the interviews in a sense the pretesting of the survey? This is not clear.

Response: Thank you for raising this point. We did the cognitive interviews to pretest the questionnaire. In the cognitive interviews, we used the pre-final version of the questionnaire (after passing the forward translation, backward translation, and expert revision) to assess the readability and understanding of the questionnaire before being widely distributed. The population were 5 persons from each selected country, with mean age (31±9), 50% males, and 25% were healthcare workers. Data was collected by the representative researcher from each country. The interview took about 15 minutes. In cognitive interviews, we don’t analyze results, we only noted respondent comments during the interview, we followed all the steps mentioned by Kristen Miller and Stephanie Willson in their book’’ Cognitive interviewing methodology’’, 2014. In the method section, we described in details the difficulties we faced with some items of the questionnaire during cognitive interviews [P: 8, L:221-240]. 

* I am not sure if I understand whether the authors claim to have used a random sample or not. They state that the countries selected were random. It is hard to believe that because to do such a multi-country study one needs to have established collaborative relationships with researchers from that country. In addition, the survey was distributed via social media, whatsapp etc. I guess, one person was forwarding the email to another person, correct? In other words, this was convenience sampling and not random. A random sample would have been if a huge list of potential participants was given to the researchers and they chose randomly 600 of those and sent them the survey via email etc. How were the participants recruited? It is not clear.

Response: We thank you for raising this point. We clarify the sampling technique in the method section [P:6, L:169-175]. We are a group of Arab researchers (48 researchers) implemented this study as a part of a large multinational project aiming at assessing the psychological antecedents against COVID-19 vaccine among Arab population living in (Egypt, Sudan, Libya , Tunisia, Morocco, Mauritania, Jordan, Palestine, Lebanon, United Arab of Emirates (UAE), Saudi Arabia, Oman, Kuwait, and Yemen). After translating the questionnaire, it was revised by all researchers in the project from the 14 countries. Then, we randomly selected four countries (Egypt, UAE, Libya, and Saudi Arabia) from this list for the aim of this study. A representative researcher from each selected country was responsible for collecting the data from the population living in this country. The representative researcher submitted the questionnaire online via social media groups and emails from his country. People were invited to agree to fill in the questionnaire.

* Under data management and psychometric analysis the authors write that “qualitative data were presented with percent and frequency”. Qualitative data refers to text and comments. Did the authors mean something else?

Response: Qualitative data refer to data or variables that are not numerical and could fit into categories like questions answered yes/no or sex. We now changed it to qualitative variables instead of qualitative data to make it clearer.

* Was the survey written in classical or formative Arabic language? Were there any difficulties in understanding some of the words even in classical Arabic given the different dialects?

Response: Thank you for noting this. We used the formal Arabic language considering simple terms to be easily understood between different people from different Arab countries. We tried to consider Arabic words that could be easily understood by different people. The final Arabic copy of the questionnaire was revised by the 48 researchers from the 14 Arab countries included in our project. In addition, during cognitive interviews we tested the questionnaire among 20 population (5 persons/ each country) to check its clarity. We faced some difficulties in explaining some terms and we mentioned this in the manuscript [P: 8, L:221-240]. 

Results:

* The results seem well written however, one can see from the demographics that the study sample is skewed towards the higher educated populations since almost 80% of the study population were at least University graduates. Another observation is that authors do not mention the ethnic group of the participants, just the country. Were all participants Arabs? This was not clear.

Response: We do agree with you about the degree of education among the population. We know that there may be selection bias, as in all online surveys. However, the percentage of university graduates showed sharp rise in several Arab countries during the last couple of years. Based on ‘’the Media Use in the Middle East, 2019 ’A seven-nation survey by Northwestern University in Qatar’ ’http://www.mideastmedia.org/ and http://www.unesco.org/new/fileadmin/MULTIMEDIA/FIELD/Beirut/video/Report.pdf.

 Also, all published studies in the Arab region to assess the COVID-19 vaccine acceptance reported similar percentages of university and postgraduates among their study population.[Ref 19-23]. We also referred to selection bias as one of the limitations to our study

All study population were Arabic because this is our focus. We now added information on the nationality of the study sample [P:12,L:337-339].

* Another observation is that more than one third of the participants had a relative who died from COVID. This could have influenced the motivation to accept the vaccine. I would suggest that one of the exclusion criteria in the recruitment process was having a relative who died from Covid because this subpopulation might have different motivations toward getting the vaccine compared to the general public because of witnessing a loved one dying from COVID. Were any exclusion criteria established?

Response: Thank you for raising this issue. As for excluding people with relative who died due to COVID, we thought that this will affect the generalizability of our data as those represents a good percentage in our societies. We now checked the effect of having a relative who died due to COVID-19 on the intention to get the vaccine by doing a regression analysis to determine the predictors of intention to get the vaccine. Based on our results, this variable has not significantly affected the people intention to be vaccinated [P: 12, L: 352-358]. For the exclusion criteria in our study, we considered only in our study Adult (18 years and above) Arabic speaking population from Arab countries are included in the study [P: 9, L:247-248].

Discussion

* The authors claim that the results of the study are representative of the Arab region. This is true up to an extent, countries like Lebanon, Jordan and Syria, the Levant region are not included. Are those represented as well?

Response: We thank you for raising this point. We now added information on the nationality of the study population. Among them, there were 18% who were originally from the Levant region(Jordan, Syria, and Lebanon). We also modified the sentence to remove the word representative. 

*The discussion was very shallow and it is focused mostly on the psychological interpretations of the results. The authors used vague terms like “ the individual” or the “people”. Well these people are Arab people. I suggest that the author interpret the result through the lenses of Arab culture and Islamic religion. For instance it is well known, that Arab societies are collectivist societies, how does that relate to the results? What about the trust in God or fate, that if someone gets sick it was meant to happen? Or that disease is perhaps punishment from God? How can public health professionals who plan interventions to promote vaccine acceptance in these countries can use the results of this study?

Response: We thank you for your suggestions. We tried to enrich the discussion considering your advice. For the effect of the collectivist Arabic societies on our results, we have not tested this as it is out of our scope in this manuscript. In the other project where we will test the vaccine hesitancy among the general population living in Arab region, we consider all obstacles that may have an impact on the decision to take or not to take the vaccine.

* How do the authors interpret the results based on the fact that almost a third of the sample had someone who had died from COVID? Do they think that might have influenced the way the participants answered the questions or not?

* What are the study limitations?

Response: Thank you for noting this. We now tested the effect of having a history of someone who died from COVID-19 on intention to get the vaccine and we found that it is not significantly predicting the decision. Moreover, based on the published studies in the Arab region, the most commonly determinants for COVID-19 vaccines acceptance were age, gender, education level, confidence in the health system and safety of the COVID-19 vaccine [Ref: 19-23].

We now added a paragraph on the study strengths and limitation [P16:, L474-476 [P:17,L:477-492].

Reviewer #2: 

In general, it is a very interesting topic specially that validations scales in Arabic are very needed studies. Researchers have done good job in the statistical analysis of the study and explained their methodology very well. Few suggestions can be provided to make the paper sound even better:

Response: Thank you for your comments,

Introduction:

*The flow of the introduction can be improved paragraph1 is about the outbreak of COVID-19. Paragraph2 facing COVID-19 globally. Paragraphs 3,4 and 5 can be combined about the COVID-19 vaccine. Other paragraphs can be used to introduce: acceptance of the vaccine globally, if there are different tools used to understand vaccine psychological impact, if there any theoretical and conceptual framework used to develop the current tool, and finally stating the aim of the study.

Response: Thank you for your suggestions. We have updated the introduction as suggested. 

Method:

*The method section is too long! Is there a way to shorten it? Keep important information in the method section and extra information in the supporting document if this is possible.

Response: Thank you for your comment. Actually, we tried to present the important steps in the process of validation. Now we removed the items of the questionnaire from the method section and put it as supplementary1.

*For study tool for example, keep the sub-scales and their definitions. Categories of the subscales can be moved in the supporting document.

Response: Thank you for your advice. We did it and moved the items to supplementary table 1.

*Although the mode of the survey was mentioned, it is not clear to me how the study targeted the population of the four countries Egypt, Libya, UAE and KSA? The availability of the survey online means that the survey was open to anyone from any country? What measures or strategies researchers did to ensure targeting the specified countries mentioned? This need to be explained to remove ambiguity.

Response: We thank you for raising this point. We clarify the sampling technique in the method section [ P: 6, L:169-179]. We are a group of Arab researchers (48 researchers) implemented this study a part of a large multinational project aiming at assessing the psychological antecedents against COVID-19 vaccine among Arab population living in 14 Arab countries (Egypt, United Arab of Emirates (UAE), Saudi Arabia, Sudan, Jordan, Kuwait, Libya, Yemen, Morocco, Palestine, Lebanon, Sudan, Oman, and Mauritania). After translating the questionnaire, it was revised by all researchers in the project from the 14 countries. Then, we randomly selected four countries (Egypt, UAE, Libya, and Saudi Arabia) from this list to for the aim of this study. A representative researcher from each selected country was responsible for collecting the data from the population living in this country. The representative researcher submitted the questionnaire online via social media groups and emails from his country. People were invited to fill in the questionnaire. 

*Score interpretation: can you further clarified, and refined in a simpler way for the readers to understand.

Response: We thank you for your suggestion. We now modified it to make it more clear [P: 7, L194-202:]. In addition, we now have another accepted paper for publication based on 500 of the Arab population, in this paper we identified the different cutoff points for the subscales of 5C questionnaire. We did not cite it as it is has no doi yet.

*Translation and adaptation: how many total researchers/co-authors/ translators worked for this section? This section can be refined by stating the total number of individuals who worked on this section and then each stage of the translation should state the number of individuals who worked at that stage.

Response: Thank you for your comment. We added the total number of the persons included in this step as a first sentence, then we described in details. Two co-authors who are native English speakers were responsible for forward translation. Then, another bilingual researcher together with an Arabic translator (hired for this task from a translation office) assessed the translation and reconciled the discrepancies. After this step, other two co-authors who are bilingual back translated the questionnaire to English. The English translated copy and the original one were compared by the back translators and the first author to ensure the proper translation [P:7,L:203- 209 -P:8,L:210-212]. 

*Translation and adaptation: are the same translators who translated the tool into Arabic assessed the translation or separate individuals? Just to avoid bias.

Response: No, they are different and translation was done independently. we clarified this in details in the above question.

*Cognitive interviews: you mean piloting right? What about adding the word “pilot” since people are more familiar with the word pilot compared to cognitive interview? Or is it because it is a psychological study the term cognitive interview is preferred compared to the word piloting?

Response: Yes cognitive interview is a technique for pilot testing the questionnaire in the context of assessing its psychometric properties. We added the term to the heading in the method section [P:8,L:220].

* Do you think this paragraph: Based on the sample size recommendations of having 10 participants respond to each item for validating a questionnaire (ratio 10:1), we needed 150 participants [25]. 8 Moreover, a priori sample size calculation for Structural Equation Modelling (SEM) technique to perform confirmatory factor analysis (CFA) showed that a minimum sample of 200 is required to run CFA [26]. For that, the minimum required sample size for our analysis was 350 participants. Adult (18 years and above) Arabic speaking population is included in the study, should be labeled as a sample size calculation or study sample size?

Response: We agree with your suggestion and we modified it [P:9, L:241].

*Do you think this paragraph: The final Arabic copy of 5C scale was uploaded on Qualtrics and disseminated online via different social media platforms (Facebook, WhatsApp, emails, and Twitter) to 673 participants? The sample was recruited from four randomly selected Arabic countries (Egypt, Saudi Arabia. Libya, and United Arab of Emirates (UAE)). A total of 511 responded to the questionnaire, 89 participants chose not to complete the questionnaire. The response rate was 62.70% (422/673). Of the 422 who completed the questionnaire, we excluded 72 responses from the final analysis due to incomplete or inconsistent data. The final sample size included in our analysis was 350 participants, could be labeled as data source?

Response: Thank you for your suggestion. We modified it according to your advice. [P:9,L:249-263].

*What about adding: time frame of the study? When did the data collection start and ended? For how long the survey was open to participants?

Response: We do agree with you. The questionnaire was available online from December 14th, 2020 till January 14th, 2021. Now we added it to the method section [P: 9, L: 252].

*Do you think converting the paragraph above to “figure as a flow chart for participant recruitment”, will make it easier for the reader?

Response: We thank you for noting this. Now we added the flow chart to the result section (Figure 1).

*What about clarifying the design of the study and stating that it is a quantitative and a cross sectional study?

Response: we do agree with you and we added it [P:6, L: 169].

Discussion:

*Discussion should focus on interpreting the main results of the study no repeating the results

Response: Thank you for your suggestion. We modified the discussion and addied information on findings from other studies.

* Comparing the results of the Arab words to studies conducted elsewhere (missing?)

Response:We agree with you and now we changed the discussion.

* Strengths and limitation of the study (missing?)

Response: we have added a strength and limitation section [P16 & 17, L:474-492]

Conclusion:

*Why there is no separate conclusion to this study?

R: thank you for your advice. We added it to the manuscript [P: 17, L:493-495].

*Conclusion should summarize main findings, any policy implications of the study, any recommendations, any future research to answer or understand questions raised by this study.

Response: we agree with you and we added it [P: 16,L:466-472-25] & [P: 18,L:466-468].

Reviewer #3: 

*This manuscript is well written. However, there are some comments that need clarifications.

Response Thank you for your comment.

Introduction:

*Page 4 second paragraph: Authors need to differentiate between the acceptability and the availability of the vaccine in these listed countries. As these countries varies in their economic status and the availability/ affordability of the vaccine.

Response: We thank you for raising this important point. Most of vaccine acceptance studies were focusing on testing the people acceptance to get the vaccine . We now added a paragraph on the availability of COVID-19 vaccine in the Arab countries [P:4, L:102-112].

Methodology:

*the study design should be clearly mentioned, and the authors should make is clear that they used a mixed approach (both qualitative and quantitative). Also, the reasons behind using the mixed approach (quantitative and the qualitative).

Response: We thank you for your suggestion. Now, we added the study design under the heading ’’study design and setting’’[P: 6, L:169]. We did not use mixed method, it is only quantitative research. The term qualitative we used in the statistical analysis refers to the qualitative variables which means all variables that are not numerical as sex, ethnicity, history of COVID-19 infection, etc. We modified the word in the method section to refer to the qualitative variables. 

*Its not clear how the participants were recruited from four selected Arabic countries? how did they select the four countries randomly? (Egypt, Saudi Arabia. Libya, and United Arab of Emirates (UAE). The authors mentioned that the four Arab countries were randomly selected. it’s not clear how they selected these countries in a random basis? and why four and not more or less?

Response: This study is a part of another project aimed to assess the vaccine hesitancy among Arab population. We are a group of Arab researchers (48 researchers) from 14 Arab countries(Egypt, Sudan, Libya , Tunisia, Morocco, Mauritania, Jordan, Palestine, Lebanon, United Arab of Emirates (UAE), Saudi Arabia, Oman, Kuwait, and Yemen). In the validation study, we randomly selected 4 countries out of the 14 Arab countries and so the country selection was based on random technique. For the study people, the questionnaire was distributed online and each representative researcher was responsible to invite people from his country to participate in filling in the questionnaire through sending it via emails or distributing it on the social media platforms. Here is the link for the project to assess the vaccine hesitancy among the Arab population (https://www.researchgate.net/project/Covid-19-Vaccine-Hesitancy-in-the-Arab-Word)

*By reading the manuscript, I thought that the sampling methodology is non-proportional convenient sampling. Those included in the study are participants who are available and volunteered to participate.

Response: the study participants were invited to participate in filling the questionnaire via social media platforms and emails. Every representative researcher was responsible for circulating the questionnaire in his country to recruit the study population. Due to the COVID-19 restrictive measures, online surveys were the safest way to implement the research. We now added it to the limitations [P:17, L:480-484]

*Page 8 at the end of paragraph 1 " we excluded 72 responses from the final analysis due to incomplete or inconsistent data". Are these questionnaires distributed over selected countries? what is the percentage from each country?

Response: Yes, those questionnaires were from the four selected countries. They distributed as 33 from Egypt, 16 from Libya, 12 from Saudi Arabia, and 11 from UAE. We added it to the manuscript [P: 9, L:261-263].

*End of page 8, beginning of page 9, its not clear how the researchers measure the construct validity and how they randomly divide the sample into 2 groups one with 150 and the other with 200?

Response: The construct validity was measured through assessing the criterion related validity (concurrent, convergent and discriminant) and by doing factorial analysis following recommendation by references 33 and 34, we highlighted this in the method section. The random division of the sample was done using computer generated approach. All data of the 350 study sample were entered to SPSS program and a biostatistician added an ID number for each participant. Then, using a special function in SPSS (data… select cases… random sample of cases) the software randomly selected 150 participants for EFA and the remaining 200 were included in CFA analysis.

*The translated Arabic version of the questionnaire should be provided with the manuscript.

Response: thank you for your suggestion. We now add it to the manuscript as supplementary 2. 

Results:

*Because there is high discrepancy between the lowest and the highest age that affected the mean, its better to categorize the age and then run the analysis. Also, about one third of the participants are from Egypt whose Arabic accent and words meaning to some extent is different than other Arab countries. Also, none of the following countries are included in the sample (Jordan, Lebanon, Syria, and Palestine) who have seminaries in Arabic while they are different than other Arab countries, this might create bias in sample selection and the results.

Response: We thank you for raising this point. As for age, we already categorized it in the table 1 and now we added a comment on it in the text [P:12 L: 335-337] and also in the regression analysis, we compared between the intention to get COVID-19 vaccine between those aged less than 40 years versus those 40 years or more [P:12, L:352-358].

 We do agree with you the Arabic accent among Egyptians are different from other countries. That is why we translated the questionnaire using the formal Arabic language considering simple terms that could be understood by people from different countries. Also, the Arabic copy of the questionnaire was revised by the 48 researchers from the 14 Arab countries and they gave their feedback. We agree with you that no country from Levant region was considered in our analysis. Actually, we have a researcher representative from Jordan, Syria, and Lebanon but random selection of the included countries did not come with one of it. However, we have some people from those countries who live in the included countries. We add information on the nationality in table 1. 

Discussion

*Page 11: I disagree with the authors statement “Therefore, chosen countries in this study are good representative of the Arab region". They missed to include one representative country from: (Jordan, Lebanon, Syria, and Palestine). Authors should mention that.

R: Thank you for your comment. We removed the statement from the discussion and we described the nationality of the included study population. Among them, there were 18% from Levant region (Jordan, Lebanon, and Syria).

---

## [Decision Letter · Decision Letter 1]

30 Jun 2021

Arabic validation and cross-cultural adaptation of the 5C scale for assessment of COVID-19 vaccines psychological antecedents

PONE-D-21-03821R1

Dear Dr. Elbarazi,

We’re pleased to inform you that your manuscript has been judged scientifically suitable for publication and will be formally accepted for publication once it meets all outstanding technical requirements.

Kind regards,

Adewale L. Oyeyemi, Ph.D

Academic Editor

PLOS ONE

Additional Editor Comments (optional):

Reviewers' comments:

Reviewer's Responses to Questions

**Comments to the Author**

1. If the authors have adequately addressed your comments raised in a previous round of review and you feel that this manuscript is now acceptable for publication, you may indicate that here to bypass the “Comments to the Author” section, enter your conflict of interest statement in the “Confidential to Editor” section, and submit your "Accept" recommendation.

Reviewer #3: All comments have been addressed

2. Is the manuscript technically sound, and do the data support the conclusions?

Reviewer #3: Yes

3. Has the statistical analysis been performed appropriately and rigorously? 

Reviewer #3: Yes

4. Have the authors made all data underlying the findings in their manuscript fully available?

Reviewer #3: Yes

5. Is the manuscript presented in an intelligible fashion and written in standard English?

Reviewer #3: Yes

6. Review Comments to the Author

Reviewer #3: I would like to congratulate the authors of this manuscript who did a great job in revising the manuscript and endorsed all my comments.

7. PLOS authors have the option to publish the peer review history of their article (what does this mean?). If published, this will include your full peer review and any attached files.

Reviewer #3: **Yes: **Haleama Al Sabbah

---

## [Editor Report · Acceptance letter]

12 Aug 2021

PONE-D-21-03821R1 

Arabic validation and cross-cultural adaptation of the 5C scale for assessment of COVID-19 vaccines psychological antecedents 

Dear Dr. Elbarazi:

I'm pleased to inform you that your manuscript has been deemed suitable for publication in PLOS ONE. Congratulations! Your manuscript is now with our production department. 

Kind regards, 

on behalf of

Dr. Adewale L. Oyeyemi 

Academic Editor

PLOS ONE